# Impact of Molasses on Ruminal Volatile Fatty Acid Production and Microbiota Composition In Vitro

**DOI:** 10.3390/ani13040728

**Published:** 2023-02-17

**Authors:** A. Palmonari, A. Federiconi, D. Cavallini, C. J. Sniffen, L. Mammi, S. Turroni, F. D’Amico, P. Holder, A. Formigoni

**Affiliations:** 1DIMEVET, Dipartimento di Scienze Mediche Veterinarie, Università di Bologna, 40064 Ozzano Emilia, Italy; 2Fencrest LLC, Holderness, NH 03245, USA; 3Unit of Microbiome Science and Biotechnology, Department of Pharmacy and Biotechnology, University of Bologna, 40126 Bologna, Italy; 4Unit of Microbiomics, Department of Medical and Surgical Sciences, University of Bologna, 40138 Bologna, Italy; 5ED&F Man Liquid Products, 3 London Bridge Street, London SE1 9SG, UK

**Keywords:** molasses, VFAs, rumen microbiota, in vitro fermentation, 16S rRNA amplicon sequencing

## Abstract

**Simple Summary:**

Molasses is extensively used in ruminant nutrition. Different studies have evaluated their effects on VFA production, but no data are available on the potential changes in rumen microbiota. The aim of this study was to assess the way in which the use of molasses could have modified VFA production and the rumen microbial community in vitro. The obtained results show the ability of molasses to impact the rumen microbiota, leading to increased proportions of some peculiar bacterial families and reduced amounts of others, thus resulting in different VFA production and composition.

**Abstract:**

The aim of this study was to assess if molasses could modify VFA production and the rumen microbial community in vitro. Three beet (treatment Beet) and three cane (treatment Cane) molasses preparations were randomly selected from a variety of samples collected worldwide and incubated in vitro with rumen fluid along with a control sample (treatment CTR, in which no molasses was used). Flasks for VFA analysis were sampled at 0, 1, 2, 3, 4, 6, 8, and 24 h of each incubation. For microbiota analysis, samples from each fermentation flask after 12 and 24 h were subjected to microbial DNA extraction and V3–V4 16S rRNA gene sequencing on an Illumina MiSeq platform. Total net VFA production was higher in the beet and cane preparations than in the control (CTR) group at 24 h (33 mmol/L, 34 mmol/L, and 24.8 mmol/L, respectively), and the composition of VFAs was affected by the inclusion of molasses: acetic acid increased in the CTR group (73.5 mol%), while propionic acid increased in the beet and cane molasses (19.6 mol% and 18.6 mol%, respectively), and butyric acid increased, especially in the cane group (23.2 mol%). Molasses even influenced the composition of the rumen microbiota, and particularly the relative abundance of the most dominant family in the rumen, Prevotellaceae, which decreased compared to CTR (37.13%, 28.88%, and 49.6%, respectively). In contrast, Streptococcaceae (19.62% and 28.10% in molasses compared to 6.23% in CTR), Veillonellaceae (6.48% and 8.67% in molasses compared to 4.54% in CTR), and Fibrobacteraceae (0.90% and 0.88% in molasses compared to 0.62% in CTR) increased in the beet and cane groups compared to the CTR group. Another important finding is the lower proportion of Methanobacteriaceae following the addition of molasses compared to CTR (0.26%, 0.28%, and 0.43%, respectively). This study showed the impact of molasses in influencing VFA production and composition as a result of a modified rumen microbial composition.

## 1. Introduction

Beet and cane molasses are produced and used worldwide in animal nutrition. As reported in several studies, dietary addition of molasses could improve dry matter intake (DMI), reduce sorting, affect milk fat, fat-corrected milk (FCM), ruminal ammonia, milk urea nitrogen (MUN), fiber digestibility, and butyrate production [1]. Molasses composition is not constant and could vary among different samples. The origin of the molasses, as well as climatic conditions or industrial treatment, could cause such differences, as reported in a previous study [2]. However, the most important fraction remains sugar, which could account for up to ~50% of their content. This fraction is mainly composed of mono- and disaccharides of hexoses, such as sucrose, glucose, and fructose, which are rapidly fermented in the rumen at a high digestion rate that could reach values of 50–60%/h [3]. It is generally assumed that, as a consequence, the addition of molasses to the diet would impact rumen VFA production and composition [4]. Overall, the reported data show the effects of molasses in reducing the amount of acetate produced, as well as an increased concentration of propionate and butyrate, in particular. This aspect is also related to a greater volume of gas produced due to the rapid availability of the substrate. Rapidly fermentable carbohydrates speed up fermentations, being an excellent energy source for microbial metabolism [5]. Ruminal microbes carry out the degradation and fermentation of plant material such as starch, cellulose, and soluble sugars, which result in the conversion into digestible compounds, and it is influenced mainly by the diet [6]. According to Palmonari et al. [3], the most representative sugar in beet and cane molasses is sucrose which has a higher digestibility than other sugars, and this rapid digestion is related to increased DMI and improved rumen microbial protein synthesis. Sucrose is a disaccharide composed of glucose and fructose, major substrates for rumen microbial fermentations. The dietary addition of molasses could improve fiber digestibility and enhance ruminal biohydrogenation of fatty acids, reducing milk fat depression development in cows [7]. However, despite the characterization of VFA and gas production, no studies have been carried out on the impact of molasses addition to the rumen microbial community in vitro. In other words, other studies evaluated the effects of molasses on the differences in end products of microbial fermentations but not on the microbial community itself. The aim of the current study was to assess the way in which molasses might modify the rumen microbiota composition and VFA production during in vitro incubations of rumen fluid.

## 2. Materials and Methods

### 2.1. In Vitro Incubations and VFAs

The procedure performed for the in vitro incubations was described in previous works [2,3]. Basically, three beet and three cane molasses were randomly selected from a variety of samples collected worldwide. Chosen cane molasses came from Central America, Asia, and Europe, while beet molasses came from Europe, North America, and Africa. Samples were analyzed as described [2], and their compositions are reported in Table 1. The six samples were tested in two different in vitro incubations with rumen fluid. For the inoculum, three lactating Holstein cows were selected as donors based on similar BW, parity, DIM, milk production, and milk composition (SCC, fat and protein, lactose, and urea). Animals were milked twice a day. Donor cows were fed a hay-based diet containing alfalfa hay (48% aNDFom), grass hay (50% aNDFom), and corn grain (65% starch). Rumen fluids were sampled via an esophageal probe, pouring off the first volume collected to avoid saliva or mucous contamination, and immediately placed in a thermostatic bottle. This method would allow the recovery of the liquid phase, while only fine particles of the solid one could be sampled. However, such characteristic does not represent an issue for the current study since treatments do not contain traces of fibrous material. Sampling was conducted at 0300 h after feeding. Rumen content was filtered through 2 layers of cheesecloth under constant O_2_-free CO_2_. Once filtered, an equal amount of each liquor collected was sampled and equally mixed with the others. A final volume of 1200 mL for each incubation was obtained, containing 400 mL of rumen fluid collected from each cow. In vitro fermentations were conducted following the procedure described by Palmonari et al. [3], with few modifications. To ensure enough residue was collected at any given time point, the volume and amount of sample weighed into each incubation flask was tripled compared to the original procedure. This adjustment was able to provide an equal amount of sample in each flask. The final content of each flask was 1.50 g of sample (90% DM diet fed to animals + 10% DM molasses to test), 120 mL of buffer solution as described by Goering and Van Soest [8], and 30 mL of rumen fluid. Three replicates per each sample of the two treatments and three replicates of the control (CTR, no molasses addition) were incubated per time point. Flasks were placed in a heated (39.3 °C) water bath under CO_2_ positive pressure to ensure anaerobiosis and mixed using a magnetic stirrer during the fermentative process. Briefly, each flask was sealed with a screw cap equipped with two separate valves, one “in” and one “out”. CO_2_ was insufflated through the “in” valve, and the gas flowed from the “out” valve. The two different incubations lasted for 24 h, and two days elapsed between them.

Flasks for VFA analysis were sampled at 0, 1, 2, 3, 4, 6, 8, and 24 h for each incubation. Concentrations were determined by gas chromatography [9]. Briefly, VFAs were separated using a Fisons HRGC MEGA 2 series 8560 (Fison Instruments, Glasgow, UK) with a flame ionization detector (Fison Instruments, Glasgow, UK)) on a 2 m glass column (inner diameter, 3 mm) of 10% SP-1000 + 1% H3PO4 on 100/120 ChromosorbWAW (Tehnokroma Analitica S.A., Sant Cugat del Vallès, Spain) with nitrogen as the carrier gas. The temperature of the injector and detector was 200 °C, and the oven temperature was 155 °C. The internal standard adopted was 2-ethylbutyric acid (Sigma Aldrich, Taufkirchen, Germany).

### 2.2. Microbial DNA Extraction and 16S rRNA Amplicon Sequencing

Immediately after collection, 45 mL of rumen fluid was sampled from the batch previously described to represent the time 0 (T0) sample. Once the incubation started, 45 mL were collected from each flask after 12 and 24 h, then frozen at −80 °C. The procedure for microbial DNA extraction followed the protocol described by Stevenson and Weimer [10]. Briefly, samples were filtered through 3 layers of sterile cheesecloth and then centrifuged at 10,000× *g* for 30 min (4 °C). After this, the supernatant was poured off, and the pellet was resuspended in 3 mL of cold extraction buffer, then 1 mL was transferred to a 2 mL microcentrifuge tube with the addition of microbeads, sodium dodecyl sulfate (SDS) and equilibrated phenol. Cell lysis was achieved by two runs of bead disruption (5 min each, 4 °C), with a heat shock (60 °C for 10 min) step between the two. The supernatant and phenol phases were separated by microcentrifugation (12,000× *g*, 10 min). After this step, the aqueous phase was extracted twice more with 500 μL phenol (pH 8.0), twice with 500 μL phenol/chloroform, and twice with 500 μL chloroform. The final supernatant was combined with 0.1 vol of 3.0 M sodium acetate and precipitated with 0.6 vol of isopropanol. The final pellet was cleaned with ethanol (70%) and centrifuged. The cleaning steps were repeated three times, and then the pellet was dried at room temperature for ∼45 min. Once dried, the pellet was resuspended in TE buffer, then stored at −80 °C prior to library preparation. DNA concentration and quality were assessed using a NanoDrop ND-1000 spectrophotometer (NanoDrop Technologies, Wilmington, DE, USA).

For 16S rRNA amplicon sequencing, the protocol described by Turroni et al. [11] was followed. Briefly, the V3-V4 hypervariable regions of the 16S rRNA gene were amplified using the 341F and 805R primers with added Illumina (Illumina, San Diego, CA, USA) adapter overhang sequences, as previously described. PCR products were purified with a magnetic bead-based clean-up system (Agencourt AMPure XP; Beckman Coulter, Brea, CA, USA). Indexed libraries were prepared by limited-cycle PCR using Nextera technology, further cleaned up with AMPure XP magnetic beads (Beckman Coulter), and pooled at equimolar concentration. The sample pool was denatured with 0.2 N NaOH and diluted to 5 pM with 20% PhiX control. Sequencing was performed on an Illumina MiSeq platform using a 2 × 250 bp paired-end protocol according to the manufacturer’s instructions.

### 2.3. Bioinformatic and Statistical Analysis

Raw sequences were processed using a pipeline combining PANDAseq [12], QIIME 2 ([13]), and DADA2 [14]. High-quality reads were clustered into high-resolution amplicon sequence variants (ASVs), and the taxonomy was assigned using SILVA as the reference database. Singletons and chimeras were removed during sequence processing. ASV tables were collapsed at all phylogenetic levels, from phylum to genus. Alpha diversity was computed using the number of observed ASVs, and Faith’s phylogenetic diversity metrics. Beta diversity was estimated by computing weighted and unweighted UniFrac distances, which were used as input for principal coordinates analysis (PCoA). Statistical analysis was performed in R 3.3.2 using R studio 1.0.136. Data separation in the PCoA was tested using a permutation test with pseudo-F ratios (function Adonis in the “vegan” package of R). Significant differences in alpha diversity and relative taxon abundances were assessed by the Kruskal–Wallis test, followed by the post-hoc Wilcoxon test, paired or unpaired as needed. *p* < 0.05 was considered statistically significant. Finally, to evaluate any possible correlation among VFAs and microbial families, Pearson correlation coefficients were calculated using the software JMP.

For the VFAs, the first comparison was made to evaluate any difference between the two different incubations. The ANOVA procedure of the software JMP (version 17.0 pro, Statistical Analysis Systems Institute Inc., Cary, NC, USA) was adopted. Once run, no differences were observed across the two fermentations; thus, the model was applied again to evaluate any possible difference between samples. Treatment (cane, beet, and CTR), timepoint, and their interaction were considered fixed effects, while replicates were treated as random effects. Means were then compared using the Tukey post-hoc test, and the significance was set at *p* < 0.05. ANOVA analysis was also performed to estimate potential differences between the two treatments (beet vs. cane molasses) in terms of their chemical composition.

## 3. Results

The two treatments showed significant differences in terms of chemical composition (Table 1). In particular, while sucrose was not statistically different across samples, glucose and fructose were higher in cane compared to beet molasses (glucose, 4.57 vs. 0.07%, and fructose, 7.65 vs. 0.15%, on average for cane and beet, respectively; *p* < 0.01). Raffinose showed opposite results, being higher on average in beet compared to cane (0.89 vs. 0.03% in beet and cane, respectively; *p* < 0.01). Even other components resulted differently: starch was higher in cane compared to beet, while crude protein was the opposite. Interestingly, sulfates, phosphates, and nitrates were also higher in cane compared to beet molasses (*p* < 0.05).

### 3.1. VFAs

The total VFA production is reported in Table 2, expressed as mmol/L.

Net average values were similar among groups after up to 3 h of in vitro fermentation (T3), while for T4, T6, T8, and T24, VFAs produced in the beet or cane group were significantly higher than in the CTR (beet: 13.7, 23.1, and 33.0 mmol/L; cane: 13.6, 24.3, and 34.0 mmol/L; CTR: 8.7, 9.4, and 24.8 mmol/L at T6, T8, and T24, respectively; *p* < 0.01). In addition to a higher concentration of VFAs produced at longer time points, the mol % composition also showed significant differences among treatments and for the treatment x time interaction (Table 3). Acetic acid was higher in CTR than beet and cane (beet: 61.8, 63.5, and 58.2%; cane: 59.2, 60.0, and 57.0%; CTR: 70.7, 71.8, and 73.5% at T1, T4, and T24, respectively; *p* < 0.01), while propionic acid significantly increased in treatments with molasses addition compared to CTR (beet: 23.2, 22.2, and 19.6%; cane: 21.3, 21.4, and 18.6%; CTR: 17.8, 17.5, and 14.2% at T1, T4, and T24, respectively; *p* < 0.01). Butyric acid also displayed higher values in the molasses groups, cane in particular (beet: 12.5, 11.7, and 21.9%; cane: 15.7, 15.7, and 23.2%; CTR: 8.4, 8.2, and 8.6% at T1, T4, and T24, respectively; *p* < 0.01). Moreover, a tendency was observed considering T24 against all the other time points. In particular, T24 values were lower for both acetic and propionic acid and higher for butyric acid. Iso-butyric, iso-valeric, and valeric showed no differences among treatments, remaining similar even at different time points.

### 3.2. Rumen Microbiota Composition

The 16S rRNA amplicon sequencing of 36 samples yielded a total of 1,440,209 high-quality reads (mean ± SD, 40,006 ± 20,920; range, 8136–67,785). Good coverage ranged from 97.3% to 100%, indicating that nearly the full extent of bacterial diversity was captured. Alpha diversity was significantly higher in CTR than in beet or cane molasses (*p* < 0.05) (Figure 1A). Similarly, CTR was significantly segregated from other groups in the PCoA based on unweighted UniFrac distances (*p* = 0.01) (Figure 1B), suggesting that compositional differences were mainly related to non-abundant members of the microbiota. Taxonomic analysis showed several compositional differences between beet molasses, cane molasses, and CTR in rumen bacteria family composition. The most abundant family was Prevotellaceae, as expected, which had a higher relative abundance in CTR (49.68%) compared to beet and cane molasses (37.13% and 28.88%, respectively; *p* < 0.01). On the other hand, Ruminococcaceae, another important family in the rumen (about 13% on average in our dataset), showed no significant differences among treatments and over time. As for other families, the Lachnospiraceae family was higher in cows fed with beet molasses compared to cane molasses and CTR (11.08%, 9.05%, and 9.12%, respectively; *p* < 0.05). Instead, the Streptococcaceae family had a higher relative abundance both in beet and cane compared to CTR (19,62%, 28.10%, and 6.23%, respectively; *p* < 0.01). Additionally, the Veillonellaceae family was higher in both types: beet and cane molasses had average relative abundances of 6.48% and 8.67%, respectively, while the CTR result was lower (4.54%; *p* < 0.05). Pseudomonadaceae had a lower relative abundance in beet and cane molasses as well (0.06% and 0.01%, respectively; *p* < 0.05) compared to the CTR group (0.13%). As shown in Table 4, Succinivibrionaceae showed a difference between groups: it has a lower relative abundance in beet molasses (0.71%) and a higher relative abundance (2.02%) in cane molasses compared to CTR (1.28%; *p* < 0.05). An important cellulolytic family is Fibrobacteriaceae, which was higher in both beet and cane molasses (0.90% and 0.88%, respectively; *p* < 0.05) compared to CTR. The lactic acid producer, Bifidobacteriaceae, was higher in beet but not in cane or CTR (0.49%, 0.16%, and 0.13%, respectively; *p* < 0.05). An important observation was the lower relative abundance of methane producer Methanobacteriaceae family in molasses treatment (0.26% and 0.28% compared to 0.43% of CTR; *p* < 0.01).

Similar differences were observed at the genus level (Table 5). Among the numerous genera identified with the sequencing process, several of them were of interest and showed statistical differences between treatments. The relative abundance of *Metanobrevibacter* was higher in CTR compared to beet and cane molasses (0.29, 0.15, 0.19%, for CTR, beet, and cane, respectively; *p* < 0.01). The most represented genus was *Prevotella1*, which showed differences among treatments, being higher in CTR compared to the other two (34.7, 25.5, and 16.8% in CTR, beet, and cane, respectively; *p* < 0.05). Genus *Streptococcus* acted differently, with higher relative abundances in beet and cane compared to CTR (16.2, 26.3, and 5.4% in beet, cane, and CTR, respectively; *p* < 0.01), with the highest values in cane. A similar pattern resulted for the genera *Butyrivibrio* (2.99, 2.08, and 1.63% in beet, cane, and CTR, respectively; *p* < 0.01) and *Selenomonas* (1.26, 1.01, and 0.36% in beet, cane, and CTR, respectively; *p* < 0.01). *Succiniclasticum* had a higher relative abundance in CTR (3.53, 2.81, and 5.94% in beet, cane, and CTR, respectively; *p* < 0.01), as did genus *Ruminococcus* (4.77, 4.51, and 7.38% in beet, cane, and CTR, respectively; *p* < 0.01).

The Pearson’s coefficients calculated to evaluate possible correlations among bacteria families and VFA compositions showed interesting results (Table 6). Acetic acid had a positive correlation with the genera *Ruminococcus* (0.52, *p* < 0.01), *Succiniclasticum* (0.50, *p* < 0.01), and *Fibrobacter* (0.36), while it showed a negative correlation with *Streptococcus* (−0.47, *p* < 0.01), *Clostridium* (−0.31), and *Butyrivibrio* (−0.28). Propionic acid was positively correlated with the Prevotellaceae and Veilonellaceae families (0.41 and 0.31, respectively) and the genus *Selenomonas* (0.34), while it was negatively correlated with the genera *Pseudomonas* (−0.41), *Ruminococcus* (−0.33), and *Acinetobacter* (−0.47). Iso-butyric showed positive correlations with the genera *Pseudomonas* (0.37), *Succinivibrio* (0.37), and *Acinetobacter* (0.28) and negative ones with *Butyrivibrio* (−0.44, *p* < 0.01), *Streptococcus* (−0.37), and *Selenomonas* (−0.29). On the contrary, both *Butyrivibrio* and *Streptococcus* genera were positively correlated with butyric acid (0.56 and 0.48, respectively, *p* < 0.01), along with *Clostridium* (0.36), while *Prevotella1* (−0.53, *p* < 0.01), *Desulfovibrio* (−0.50), and *Ruminococcus* (−0.44) were negatively correlated. Regarding Iso-valeric acid, a positive correlation was observed with *Pseudomonas* (0.42), *Sphaerochaeta* (0.34), and *Mogibacterium* (0.34), while a negative one was observed with *Selenomonas* (−0.47), *Streptococcus* (−0.35), and *Anaerovibrio* (−0.28). The last evaluated VFA was valeric acid, which showed a positive correlation with the genera *Streptococcus* (0.31), *Pseudomonas* (0.26), and *Clostridium* (0.25) and a negative one with *Succiniclasticum* (−0.52, *p* < 0.01), *Treponema* (−0.48, *p* < 0.01), and *Ruminococcus* (−0.42, *p* < 0.01).

## 4. Discussion

The dietary addition of cane and beet molasses in dairy cows’ diets influenced VFA production. Other authors previously observed different molar proportions among acetate, butyrate, and propionate when molasses or simple sugars were added to the diet [4,15,16]. In our study, a shift to butyrate and propionate instead of acetate was observed, as well as a higher amount of VFAs produced. Given this known effect, the impact of molasses on rumen microbial composition was observed in this study. The rumen microbiota is involved in the degradation of plant compounds, and any modification of the diet affects its activity and has a cascading impact on host performance. According to Wei et al. [17], a high abundance of soluble sugars in cows’ diets influences the levels of carbohydrates and proteins available in the rumen and increases VFA synthesis, as observed in the present study, which has a potential effect on milk production. Broderick et al. [1] demonstrated positive effects of molasses on DMI, milk fat, FCM, ruminal ammonia, MUN, and fiber digestibility. The chemical composition of molasses showed that sucrose is the most represented in both beet and cane, although beet molasses had a numerically higher sucrose concentration. The differences observed in the two molasses treatments, in terms of chemical composition, would have impacted higher or lower relative abundance of specific taxa. Sucrose is composed of glucose and fructose, which represent important substrates for microbial fermentation [2]. In our study, a higher relative abundance of Lachnospiraceae, Bifidobacteriaceae, and Erysipelotrichaceae was observed, in particular, in beet molasses. These families are mainly sucrose consumers, being also able, to a certain degree, to utilize raffinose. Both of these sugars were more abundant in beet molasses compared to cane. A high sugar concentration certainly influences saccharolytic microorganisms, such as lactic acid producers.

In our trial, Streptococcaceae increased in beet and cane molasses groups compared to CTR, especially in cane. This family is predominantly a mono-saccharides consumer, and cane molasses are higher in glucose and fructose compared to beet. Even at the genus level, *Streptococcus* showed a higher relative abundance in bot cane and beet molasses treatments, with the highest values observed in cane. Microorganisms belonging to this family produce lactic acid as the main end product of carbohydrate fermentation [18]. Streptococcaceae utilizes fermentable carbohydrates to produce lactic acid as the primary end product [19], and *Streptococcus bovis*, a member of the Streptococcaceae, has been proposed as the major lactate producer in the rumen [20]. Interestingly, the increase in those families was coupled with a higher abundance of lactic-acid utilizer bacteria such as the Veillonellaceae family. Those bacteria would prevent the decrease in rumen pH by fermenting lactic acid in other organic compounds, such as propionic or valeric acid, with beneficial effects on reducing the risk of rumen acidosis. Additionally, sucrose is the most rapidly fermented compound in molasses, and its rapid digestion is related to improved rumen microbial protein synthesis. However, feeding a high-sugar diet would require a proper nitrogen balance to avoid any decrease in N utilization or an increase in N retention [21,22]. The Prevotellaceae family utilizes a broad range of substrates, including peptides, proteins, monosaccharides, and plant polysaccharides [23]. Interestingly, in the current trial, this family showed a lower concentration in beet and cane molasses compared to the CTR, suggesting a significant impact of molasses in promoting certain bacterial populations and depressing others, such as Preovotellaceae. The specific composition of molasses, even in terms of less represented components, could be related to these effects. Zhao and colleagues [24] observed that the presence of sulfates and phosphates in the medium would negatively affect the abundance of Prevotellaceae. These compounds, also contained in molasses, could have generated such a decrease. Lachnospiraceae family showed a different trend: it was higher in beet molasses but not in cane or CTR. Lachnospiraceae are mainly butyrate producers, and a higher production of butyric acid was observed in our study when molasses was added to the fermentation. The genus *Butyrivibrio* belongs to this family, and the genus-level analysis confirmed a higher relative abundance of this specific taxon in molasses treatments. Similar results were discussed by Broderick et al. [1], reporting that a high-sugar diet increases butyrate production in the rumen. Butyrate is important for rumen health and for its capability to down-regulate lactic-acid concentrations [23]. The Succinivibrionaceae family was higher in cane molasses, while they acted oppositely in beet molasses. Such an outcome could be related to the different compositions of these molasses in terms of simple carbohydrates, such as glucose and fructose: these bacteria, indeed, utilize soluble sugars as major substrates to produce acetate, formate, and lactate. Fibrobacteraceae are plant cell wall fermenters and showed a higher abundance in beet and cane molasses compared to CTR.

Molasses are composed of several sugars, one of which is arabinose, which can be found in highly digestible fibrous compounds containing pectin or hemicellulose [3]. Moreover, according to Martel et al. [7], molasses could improve fiber digestibility via their capability to stimulate fiber-digesting ruminal bacteria. However, no differences were observed among treatments on the Ruminococcaceae family, which contains several predominant ruminal cellulolytic bacteria. On the contrary, the genus *Ruminococcus* resulted higher in the CTR group compared to the beet and cane groups, suggesting that such bacteria do not compete for substrates contained in molasses against other sugar fermenters whose relative abundance was higher in the beet and cane treatments. Pseudomonadaceae is an aerobic bacteria family which predominantly utilizes organic acids rather than glucose for its growth [25]: in our trial, they were lower both in beet and cane molasses than the CTR group, with the lowest concentration observed in the cane group. Methanobacteriaceae, and as well the genus *Metanobrevibacter*, had a lower concentration in beet and cane molasses compared to CTR. Methane production is influenced by dietary carbohydrate sources and VFA profiles in the rumen. Molasses fermentation increased the butyrate and decreased acetate production. This decrease affects the concentration of H_2_ in the rumen, which has a key role in methane production, thus reducing the availability of hydrogen to synthesize methane [7,26].

The data obtained by correlation calculations showed interesting results. The positive correlation between acetate and *Ruminococcus* could have been expected since some of the major rumen cellulolytic bacteria belong to this genus, such as *R. albus* or *R. flavefaciens*, which produce acetic acid as a major end product. A similar observation could be performed for the correlation between butyric acid and *Butyrivibrio*. *Streptococcus* was positively correlated to butyric acid and negatively to acetic acid. On the other hand, the amount of correlation, both positive and negative, associated with Provotellaceae underlines the metabolic versatility of this taxon. It was negatively correlated with butyric acid, while positively correlated with acetic acid and propionic acid. As reported in another study [27], *Prevotella* plays a role in the lignocellulose fermentation process, as well as other different carbohydrates or nitrogen compounds. A very interesting characteristic is related to its fermentation end product, which mainly is propionic acid instead of acetic acid, as indeed observed in cellulolytic bacteria. This different metabolic pattern could offer interesting possibilities for reducing methane emissions related to fiber degradation.

## 5. Conclusions

In conclusion, the results obtained in this study showed that molasses impacted VFA production and proportions and had a major influence on the rumen microbial community. Beet and cane are very well digested, thus generating an amount of fermentation end products not so different among molasses but different from the control. Thus, the differences in microbial composition could be related to the whole molasses composition. However, few or no data are present in the literature about the shifts in microbial composition due to molasses addition. In this study, molasses impacted several bacteria families, promoting or decreasing their relative abundance. Our study suggests an important role of such feedstuff in driving fermentation patterns. Thus, since available data are few, and the number of incubations in the present study was only two, future studies would be highly recommended to improve the understanding of new possible strategies for molasses dietary inclusion.

## Figures and Tables

**Figure 1 animals-13-00728-f001:**
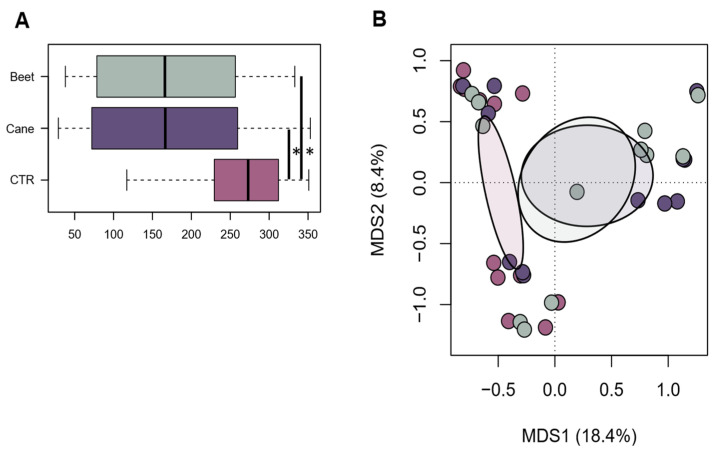
Alpha and beta diversity of the rumen microbial community in beet, cane, and CTR groups. (**A**) Boxplots showing the distribution of alpha diversity, according to the number of observed ASVs, in beet, cane, and CTR groups. Wilcoxon test, * *p* < 0.05. (**B**) Principal coordinates analysis of beta diversity, based on unweighted UniFrac distances, of all samples. A significant separation was found between CTR and other groups (PERMANOVA, *p* = 0.01). Same color code as in (**A**).

**Table 1 animals-13-00728-t001:** Molasses composition. Values are expressed as % DM unless differently specified.

		Beet		Cane			
Measure	1	2	3	1	2	3	*p*-Value
Dry Matter	76.7	79.4	78.4	76.6	78.5	76.7	0.91
Moisture	23.3	20.6	21.6	23.4	21.5	23.4	0.93
Sucrose	66.1	60.3	60.4	49.3	55.2	43.5	0.11
Glucose	0.02	0.05	0.15	5.97	1.99	5.74	<0.01
Fructose	0.05	0.12	0.28	10.02	5.03	7.9	<0.01
Raffinose	2.18	0.28	0.23	0.03	0.02	0.03	<0.01
Galactose	0.03	0.03	0.03	0.04	0.04	0.04	0.74
Arabinose	-	0.02	-	-	-	0.01	0.96
Xylose	0.01	-	-	-	-	-	-
Starch	0.09	0.03	0.1	0.54	0.18	0.32	<0.05
Levans	0.5	0.67	0.41	0.81	0.83	1.21	0.09
Destrans	0.07	0.09	0.06	0.63	1.42	0.31	<0.05
Arabinans	0.03	0.05	0.05	0.15	0.25	0.19	<0.05
Aconitic Acid	-	-	-	0.37	1.25	3.78	<0.05
Lactic Acid	3.34	3.67	6.91	3.34	6.43	12.8	0.13
Malic Acid	0.02	0.11	0.13	0.03	0.11	0.21	0.72
Citric Acid	0.39	0.38	0.15	0.08	0.13	0.19	0.81
Pyrocarbonic Acid	3.1	2.96	2.59	0.18	0.29	0.2	<0.01
Oxalic Acid	0.04	0.04	0.03	0.05	0.05	0.04	0.96
Glycolic Acid	0.22	0.26	0.23	-	-	-	<0.05
Acetic Acid	0.59	0.28	0.49	0.23	0.29	0.2	0.88
CP	12.8	14.6	12.5	7.7	8.8	6.0	<0.05
Ash	10.3	11.9	13.3	12.9	12.8	12.2	0.79
Ca	1.24	0.06	0.54	1.43	1.3	1.55	0.13
Mg	0.03	0.02	0.02	0.57	0.33	0.58	0.08
Na	0.32	1.06	0.36	0.01	0.02	0.03	<0.05
K	2.39	4.93	1.07	0.5	1.37	2.17	0.23
Sulfates	0.17	0.4	0.38	1.69	2.89	1.93	<0.05
Sulfur	0.06	0.13	0.13	0.56	0.96	0.64	<0.05
Phosphates	0.58	1.23	1.33	2.72	2.55	2.45	<0.05
Nitrates, mg/kg	35	36	18	211	784	688	<0.01
Chlorides, mg/kg	4450	797	5610	14	39	0.5	<0.01
DCAD ^1^, meq/100 g	47	129	18	−17	−20	13	<0.01

^1^ = Dietary cation–anion difference, calculated as DCAD, meq/100 g = (K, % DM/0.039 + Na, % DM/0.023)—(Cl, % DM/0.0355 + S, % DM/0.016).

**Table 2 animals-13-00728-t002:** Averaged total VFAs produced during in vitro incubations, expressed in mmol/L.

Treatment ^1^					
Time	Beet	Cane	CTR	SEM	*p*-Value
1 h	2.3	2.5	2.7	0.11	0.98
2 h	4.9	7.4	4.2	0.89	0.21
3 h	6.3	8.5	4.7	1.21	0.13
4 h	8.7	7.8	5.9	1.17	<0.05
6 h	13.7	13.6	8.7	1.58	<0.01
8 h	23.1	24.3	9.4	1.84	<0.01
24 h	33	34	24.8	2.31	<0.01

^1^ Beet = beet molasses addition; Cane = cane molasses addition; CTR = no molasses addition.

**Table 3 animals-13-00728-t003:** Averaged molar percentage of single VFAs produced by treatments during in vitro incubations.

	Treatment ^1^				
VFA, mol %	Beet	Cane	CTR	SEM	*p*-Value
Acetic					
0 h	68.8	68.8	68.8	-	-
1 h	61.8 ^b^	59.2 ^b^	70.7 ^a^	4.91	<0.01
4 h	63.5 ^b^	60.0 ^b^	71.8 ^a^	4.35	<0.01
24 h	58.2 ^b^	57.0 ^b^	73.5 ^a^	4.83	<0.01
Propionic					
0 h	18.9	18.9	18.9	-	-
1 h	23.2 ^a^	21.3 ^a^	17.8 ^b^	2.34	<0.01
4 h	22.2 ^a^	21.4 ^a^	17.5 ^b^	2.58	<0.01
24 h	19.6 ^a^	18.6 ^a^	14.2 ^b^	2.19	<0.01
Iso-Butyric					
0 h	0.48	0.48	0.98	-	-
1 h	0.40	0.72	0.84	0.47	0.91
4 h	0.41	0.55	0.76	0.35	0.74
24 h	0.73	0.57	0.93	0.33	0.86
Butyric					
0 h	8.51	8.51	8.51	-	-
1 h	12.52 ^a^	15.71 ^a^	8.42 ^b^	2.21	<0.01
4 h	11.72 ^a^	15.75 ^a^	8.25 ^b^	2.43	<0.01
24 h	21.95 ^a^	23.26 ^a^	8.67 ^b^	2.54	<0.01
Iso-Valeric					
0 h	1.1	1.1	1.1	-	-
1 h	0.72	1.24	1.1	0.61	0.33
4 h	0.73	1.07	1.2	0.62	0.42
24 h	0.86	1.01	1.4	0.15	0.98
Valeric					
0 h	1.08	1.08	1.08	-	-
1 h	1.58	1.84	1.12	0.48	0.73
4 h	1.49	1.41	0.64	0.79	0.71
24 h	1.83	1.62	1.56	0.74	0.42

^1^ Beet = beet molasses addition; Cane = cane molasses addition; CTR = no molasses addition. Least-squares means with different superscript letters within a row are significantly different (*p* < 0.05).

**Table 4 animals-13-00728-t004:** Rumen family-level microbial composition in beet, cane, and CTR groups at 24 h of incubation. For each family, the relative abundance (% of the total) is reported.

Treatment ^1^					
Family	Beet	Cane	CTR	SEM	*p*-Value
Prevotellaceae	37.13 ^AB^	28.88 ^B^	49.68 ^A^	1.51	<0.01
Streptococcaceae	19.62 ^B^	28.10 ^A^	6.23 ^C^	2.85	<0.01
Ruminococcaceae	12.76	12.04	15.19	1.20	0.48
Lachnospiraceae	11.08 ^a^	9.05 ^b^	9.12 ^b^	0.12	<0.05
Veillonellaceae	6.48 ^ab^	8.67 ^a^	4.54 ^c^	0.89	<0.05
unassigned	1.48	1.23	1.46	0.31	0.80
Erysipelotrichaceae	1.40 ^a^	0.80 ^b^	0.87 ^ab^	0.31	<0.05
TM7	1.34	1.31	1.62	0.51	0.41
Clostridiaceae	1.25	1.13	0.82	0.30	0.35
Fibrobacteriaceae	0.90 ^a^	0.88 ^ab^	0.62 ^b^	0.63	<0.05
Spirochaetaceae	0.72	0.56	1.05	0.10	0.56
Succinivibrionaceae	0.71 ^c^	2.02 ^a^	1.28 ^b^	0.63	<0.05
Coriobacteriaceae	0.61	0.31	0.28	0.80	0.26
Xanthomonadaceae	0.54	0.48	0.41	0.11	0.81
Bifidobacteriaceae	0.49 ^a^	0.16 ^b^	0.13 ^b^	0.86	<0.05
Moraxellaceae	0.40	0.32	0.57	0.22	0.86
Cyanobacteria	0.38	0.68	0.65	0.28	0.82
Fusobacteriaceae	0.32 ^a^	0.18 ^b^	0.06 ^c^	0.59	<0.05
Pirellulaceae	0.28	0.39	0.63	0.63	0.92
Vibrionaceae	0.27 ^C^	0.85 ^B^	1.50 ^A^	0.17	<0.01
Methanobacteriaceae	0.26 ^B^	0.28 ^B^	0.43 ^A^	0.83	<0.01
Anaeroplasmataceae	0.24	0.14	0.29	0.74	0.31
Pasteurellaceae	0.18 ^a^	0.03 ^b^	0.02 ^b^	0.73	<0.05
Enterobacteriaceae	0.14 ^ab^	0.21 ^a^	0.06 ^b^	0.32	<0.05
Staphylococcaceae	0.12 ^C^	0.29 ^B^	1.04 ^A^	0.40	<0.01
Desulfovibrionaceae	0.12	0.09	0.11	0.13	0.53
Victivallaceae	0.11	0.26	0.21	0.50	0.94
Sphingomonadaceae	0.09	0.20	0.13	0.56	0.20
Pseudomonadaceae	0.06 ^ab^	0.01 ^b^	0.13 ^a^	0.43	<0.05
Dethiosulfovibrionaceae	0.04	0.05	0.05	0.09	0.29
Sphaerochaetaceae	0.02	0.05	0.08	0.91	0.97
RF36	0.09	0.12	0.18	0.98	0.88

^a, b, c^ Values with different superscripts differ (*p* ≤ 0.05); ^A, B, C^ Values with different superscripts differ (*p* ≤ 0.01); ^1^ Beet = beet molasses addition; Cane = cane molasses addition; CTR = no molasses addition.

**Table 5 animals-13-00728-t005:** Rumen genus-level microbial composition in beet, cane, and CTR groups at 24 h of incubation. For each family, the relative abundance (% of the total) is reported.

Treatment ^1^					
Genus	Beet	Cane	CTR	SEM	*p*-Value
*Prevotella1*	25.5 ^B^	16.8 ^B^	34.7 ^A^	1.22	<0.01
*Streptococcus*	16.2 ^A^	26.3 ^A^	5.4 ^B^	1.85	<0.01
*Ruminococcus*	4.77 ^B^	4.51 ^B^	7.38 ^A^	1.37	<0.01
*Succiniclasticum*	3.53 ^B^	2.81 ^B^	5.94 ^A^	1.12	<0.01
*Butyrivibrio*	2.99 ^A^	2.08 ^A^	1.63 ^B^	0.78	<0.01
*Selenomonas*	1.26 ^A^	1.01 ^A^	0.36 ^B^	0.11	<0.01
*Metanobrevibacter*	0.1 ^B^	0.19 ^B^	0.29 ^A^	0.08	<0.01

^A, B^ Values with different superscripts differ (*p* ≤ 0.01); ^1^ Beet = beet molasses addition; Cane = cane molasses addition; CTR = no molasses addition.

**Table 6 animals-13-00728-t006:** Pearson’s coefficients of major correlations observed among VFAs and bacteria populations.

Pearson’s Coefficients		
VFA	Coefficient	*p*-Value
Acetic		
*Ruminococcus*	0.52	<0.01
*Succiniclasticum*	0.50	<0.01
*Fibrobacter*	0.36	n.s.
*Streptococcus*	−0.47	<0.01
*Clostridium*	−0.31	n.s.
*Butyrivibrio*	−0.28	n.s.
Propionic		
Prevotellaceae	0.41	n.s.
*Selenomonas*	0.34	n.s.
Veilonellaceae	0.31	n.s.
*Acinetobacter*	−0.47	n.s.
*Pseudomonas*	−0.41	n.s.
*Ruminococcus*	−0.33	n.s.
Iso-Butyric		
*Pseudomonas*	0.37	n.s.
*Succinivibrio*	0.37	n.s.
*Acinetobacter*	0.28	n.s.
*Butyrivibrio*	−0.44	<0.01
*Streptococcus*	−0.37	n.s.
*Selenomonas*	−0.29	n.s.
Butyric		
*Butyrivibrio*	0.56	<0.01
*Streptococcus*	0.48	<0.01
*Clostridium*	0.36	n.s.
*Prevotella1*	−0.53	<0.01
*Desulfovibrio*	−0.50	n.s.
*Ruminococcus*	−0.44	n.s.
Iso-Valeric		
*Pseudomonas*	0.42	n.s.
*Sphaerochaeta*	0.34	n.s.
*Mogibacterium*	0.34	n.s.
*Selenomonas*	−0.47	n.s.
*Streptococcus*	−0.35	n.s.
*Anaerovibrio*	−0.28	n.s.
Valeric		
*Streptococcus*	0.31	n.s.
*Pseudomonas*	0.26	n.s.
*Clostridium*	0.25	n.s.
*Succiniclasticum*	−0.52	<0.01
*Treponema*	−0.48	<0.01
*Ruminococcus*	−0.42	<0.01

## Data Availability

The data presented in this study are available on request from the corresponding author.

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
