# Peer review of "Impact of Molasses on Ruminal Volatile Fatty Acid Production and Microbiota Composition In Vitro"

_animals, 2023, doi:10.3390/ani13040728_

Round 1

Reviewer 1 Report

The authors compared fermentation end products and microbial community composition in in vitro ruminal fermentations of a mixed feed amended with cane or beet molasses, versus on the same feed without molasses addition. Clear differences were observed in the amounts of certain VFA, and in the relative abundance of several bacterial families, along with the archaea Methanobacteriaceae. Such studies have not been presented in the literature previously, so the work fills a knowledge gap, and the study appears to have been well done.  The reviewer has several major concerns regarding data presentation and interpretation. First, the VFA data do not appear to have been corrected for background levels at time-zero (see comment to Table 2 below); this issue should be easily correctable, but is necessary to accurately compare net VFA production (both amounts and relative proportions). Second, the authors should establish more detailed connections between relative abundance of certain families (Succinivibrionaceae and Lachnospiraceae in particular) and the composition of the two different types of molasses (see comment to L275 below). Because the relative abundance of Methanobacteriaceae (the dominant genus of methanogens in the rumen) was lower in the molasses supplemented groups, it is unfortunate that the authors did not measure methane production from the fermentation.

Specific comments:

L46: Suggest adding “rumen” and “in vitro fermentation” as keywords.

L86 and Table 1: There appear to be very large differences between the beet and cane molasses samples with regard to four of the component sugars (sucrose, glucose, fructose and raffinose). It would be useful to perform a means separation test between the two molasses types for each of these sugars, and state if the differences were statistically significant (say at P<0.05 or 0.01). This is important in generalizing the differences between the two molasses types, and also may allow the authors to make more definitive statements regarding the effects of the two molasses types on the abundance of specific microbial taxa.

L155-167: The authors should provide, here or in the Results section, some basic information on the sequencing metrics (average and range of number of high-quality sequences obtained for analysis, range of Good’s coverage for the samples). Also, were the number of sequences normalized for all samples to the number obtained for the sample with fewest sequences? This normalization is commonly done in microbial community analysis, although its value has been disputed (see doi: 10.1371/journal.pcbi.1003531 )

L157-159: Were singletons removed from the analysis?

Table 2: Are these millimolar values corrected for the concentration of VFA at time-zero? It seems unlikely that they were – for example, the VFA increase in CTR between 1h and 2h is only 1.5 mM, and the likely difference between 0h and 1h would probably be within this same range. The concentrations should be corrected for the zero-time value, and the means separation tests repeated with the net mM values. The corrected data should also be used to calculate the molar proportions of the net MM concentrations in Table 3.

Table 3:Does iso-valeric also include 2-methylbutyric?

L213: Here and elsewhere, use the term “were higher than” or “were lower than”, as opposed to “increased” and “decreased”, respectively. The latter terms should only be used to describe changes within a culture or treatment group (i.e., over a time course or upon mid-experiment changes in a treatment).

L217: This is a relative abundance, not a  “concentration” (an absolute measurement), and should be referred to as such.

L275: These observations may contribute to differences in bacterial community composition, but it would also be worthwhile to point out here that the higher relative abundance of Lachnospiraceae and Bifidobacteriaceae in the beet molasses group may be related to the this substrate’s higher content of sucrose and/or raffinose, while the higher relative abundance of Succinivibrionaceae in the cane molasses group may be related to the higher content of glucose and fructose.

Minor edits:

L19: Change “are” to “is”. Insert “have” ahead of “evaluated”.

L97: Here and elsewhere, subscript the “2’ in O2 and CO2 (to O2 and CO2).

Table 1: Change “xylose” to “xylose”.

L267: Change “utilized” to “utilizes”.

Author Response

The authors compared fermentation end products and microbial community composition in in vitro ruminal fermentations of a mixed feed amended with cane or beet molasses, versus on the same feed without molasses addition. Clear differences were observed in the amounts of certain VFA, and in the relative abundance of several bacterial families, along with the archaea Methanobacteriaceae. Such studies have not been presented in the literature previously, so the work fills a knowledge gap, and the study appears to have been well done.  The reviewer has several major concerns regarding data presentation and interpretation. First, the VFA data do not appear to have been corrected for background levels at time-zero (see comment to Table 2 below); this issue should be easily correctable, but is necessary to accurately compare net VFA production (both amounts and relative proportions). Second, the authors should establish more detailed connections between relative abundance of certain families (Succinivibrionaceae and Lachnospiraceae in particular) and the composition of the two different types of molasses (see comment to L275 below). Because the relative abundance of Methanobacteriaceae (the dominant genus of methanogens in the rumen) was lower in the molasses supplemented groups, it is unfortunate that the authors did not measure methane production from the fermentation.

Specific comments:

L46: Suggest adding “rumen” and “in vitro fermentation” as keywords.

AU: key words added

L86 and Table 1: There appear to be very large differences between the beet and cane molasses samples with regard to four of the component sugars (sucrose, glucose, fructose and raffinose). It would be useful to perform a means separation test between the two molasses types for each of these sugars, and state if the differences were statistically significant (say at P<0.05 or 0.01). This is important in generalizing the differences between the two molasses types, and also may allow the authors to make more definitive statements regarding the effects of the two molasses types on the abundance of specific microbial taxa.

AU: analysis added, and differences stated in the results section

L155-167: The authors should provide, here or in the Results section, some basic information on the sequencing metrics (average and range of number of high-quality sequences obtained for analysis, range of Good’s coverage for the samples).

AU: Following the Reviewer’s suggestion, in the revised version of the manuscript, we have added the requested info in the Results section, at the beginning of paragraph 3.2 “Rumen microbiota composition”.  Specifically, 16S rRNA amplicon sequencing of 36 samples yielded a total of 1,440,209 high-quality reads (mean ± SD, 40,006 ± 20,920; range, 8,136 – 67,785). Good’s coverage ranged from 97.3% to 100%, indicating that nearly the full extent of bacterial diversity was captured.

Also, were the number of sequences normalized for all samples to the number obtained for the sample with fewest sequences? This normalization is commonly done in microbial community analysis, although its value has been disputed (see doi: 10.1371/journal.pcbi.1003531 )

AU: We thank the Reviewer for raising this point and reporting this work to us.

Yes, the number of sequences for all samples was normalized to the number obtained for the sample with the least number of sequences, as commonly done in microbial community analysis and expected in the QIIME 2 pipeline.

L157-159: Were singletons removed from the analysis?

AU: Yes, singletons were removed during the sequencing process. Statement reported in the manuscript.

Table 2: Are these millimolar values corrected for the concentration of VFA at time-zero? It seems unlikely that they were – for example, the VFA increase in CTR between 1h and 2h is only 1.5 mM, and the likely difference between 0h and 1h would probably be within this same range. The concentrations should be corrected for the zero-time value, and the means separation tests repeated with the net mM values. The corrected data should also be used to calculate the molar proportions of the net MM concentrations in Table 3.

AU: the specific T0 row was added to each table. Data were not adjusted, as supposed by the reviewer, but the T0 values are identical among the treatments, since the same mix of rumen fluid was added to each bottle, and the VFAs at T0 are relative to the inoculum itself. Authors preferred to leave the total amount produced, because we think that it would better represent the fermentation process.

Table 3: Does iso-valeric also include 2-methylbutyric?

AU: yes, it does

L213: Here and elsewhere, use the term “were higher than” or “were lower than”, as opposed to “increased” and “decreased”, respectively. The latter terms should only be used to describe changes within a culture or treatment group (i.e., over a time course or upon mid-experiment changes in a treatment).

AU: terms corrected

L217: This is a relative abundance, not a “concentration” (an absolute measurement), and should be referred to as such.

AU: correction made

L275: These observations may contribute to differences in bacterial community composition, but it would also be worthwhile to point out here that the higher relative abundance of Lachnospiraceae and Bifidobacteriaceae in the beet molasses group may be related to the substrate’s higher content of sucrose and/or raffinose, while the higher relative abundance of Succinivibrionaceae in the cane molasses group may be related to the higher content of glucose and fructose.

 AU: discussion improved

Minor edits:

 L19: Change “are” to “is”. Insert “have” ahead of “evaluated”.

AU: changed

L97: Here and elsewhere, subscript the “2’ in O2 and CO2 (to O2 and CO2).

AU: changed

Table 1: Change “xylose” to “xylose”.

AU: changed

L267: Change “utilized” to “utilizes”.

AU: changed

Reviewer 2 Report

The research topic is of interest. The authors claim to have conducted the study to assess the microbiome composition following molasses supplementation. However, more microbiome data should be provided to support the purpose of the study. Hereunder my comments to the authors.

Abstract

- This section should be shortened and more concise. Please follow the word limitation according to the journal guidelines. You can easily delete the first four lines.

- You have to clearly mention the experimental treatments.  

Introduction

- L 50-51, DMI: Please define any abbreviation at its first mention. Follow that throughout the paper.

- L 52: Please follow the journals’ guidelines in citing references.

- L 52-53: Explain the reason(s) for the variation.

Materials and Methods

- L 83-84: Please provide the places from where samples are collected.

- L86-87: At least three experimental runs should be carried out in order to properly interpret and validate the in vitro results.

- L88: DMI?!

- L97 O2 and CO2: subscript

- L 103-108: It is unclear which type of the three molasses for beet and cane has been used. More clarification for the experimental treatments must be provided.

- L 108-110: Please explain briefly how you could ensure an anaerobic condition for the flasks in the water bath.

-  L159-163: There are no data for alpha and beta diversity reported in the manuscript.

- In the statistical section, I encourage the authors to conduct correlation studies between VFA and microbiome data. This would enrich the paper and help you have an in-depth discussion.

Results

- Please mention the minimum and maximum sequence reads

- What about the genus level? You can provide information on the significant findings among the groups.

-  Table 4: Please identify which time point this table refers to, 12 or 24 hours. Please arrange the families in the table in a descending manner based on their abundance.

Discussion

You need to discuss your findings more in depth. For instance, you have to explain why the Prevotellaceae family decreased in molasses groups while those groups had higher propionic acid. As you propose in L 254-258, genus data could provide new information and deeper insights into microbiome alterations. I recommend improving this section.

- L 238-239: You have not measured gas production; please delete.

Author Response

Abstract

- This section should be shortened and more concise. Please follow the word limitation according to the journal guidelines. You can easily delete the first four lines.

AU: section simplified

- You have to clearly mention the experimental treatments.  

 AU: treatments explained

Introduction

- L 50-51, DMI: Please define any abbreviation at its first mention. Follow that throughout the paper.

AU: abbreviations stated

- L 52: Please follow the journals’ guidelines in citing references.

AU: references changed according to guidelines

- L 52-53: Explain the reason(s) for the variation.

 AU: variation explained

Materials and Methods

- L 83-84: Please provide the places from where samples are collected.

AU: provided

- L86-87: At least three experimental runs should be carried out in order to properly interpret and validate the in vitro results.

AU: authors agree and understand the comment, which will be considered for the next trial

- L88: DMI?!

AU: abbreviation explained above

- L97 O2 and CO2: subscript

AU: changed

- L 103-108: It is unclear which type of the three molasses for beet and cane has been used. More clarification for the experimental treatments must be provided.

AU: sentences added. We tested all the mentioned molasses

- L 108-110: Please explain briefly how you could ensure an anaerobic condition for the flasks in the water bath.

AU: explained

-  L159-163: There are no data for alpha and beta diversity reported in the manuscript.

AU: We apologize for not including this information in the previous version of our manuscript. In full agreement with the Reviewer, we have modified the Materials and Methods and Results sections by adding data on alpha and beta diversity. We have also provided a new figure, Figure 1, which shows the required data. According to this new analysis, alpha diversity was significantly higher in CTR than in beet or cane molasses (observed number of ASVs, P < 0.05), and CTR significantly segregated from other groups in the PCoA based on unweighted UniFrac distances (P = 0.01), suggesting that compositional differences were mainly related to non-abundant members of the microbiota.

- In the statistical section, I encourage the authors to conduct correlation studies between VFA and microbiome data. This would enrich the paper and help you have an in-depth discussion.

AU: correlation made, and discussion improved.

Results

- Please mention the minimum and maximum sequence reads

AU: Data mentioned in the text. Specifically, 16S rRNA amplicon sequencing of 36 samples yielded a total of 1,440,209 high-quality reads (mean ± SD, 40,006 ± 20,920; range, 8,136 – 67,785). Good’s coverage ranged from 97.3% to 100%, indicating that nearly the full extent of bacterial diversity was captured

- What about the genus level? You can provide information on the significant findings among the groups.

AU: results for genus level analysis added.

-  Table 4: Please identify which time point this table refers to, 12 or 24 hours. Please arrange the families in the table in a descending manner based on their abundance.

 AU: time point reported in the table.

Discussion

You need to discuss your findings more in depth. For instance, you have to explain why the Prevotellaceae family decreased in molasses groups while those groups had higher propionic acid. As you propose in L 254-258, genus data could provide new information and deeper insights into microbiome alterations. I recommend improving this section.

AU: section improved. Genus level discussion implemented, and possible reason for Prevotella’s decrease given.

- L 238-239: You have not measured gas production; please delete.

 AU: gas production mention removed.

Round 2

Reviewer 1 Report

The authors have included additional analysis of their data -- particularly the microbial component, and they have expanded on the Discussion to tie the microbial differences to differences in molasses composition.  However, some issues still need to be addressed.

            In particular, the reviewer is still not convinced that the VFA data (Table 2) are properly presented and analyzed. The key point is that the net amount of VFA, rather than the amount in the vial, is what needs to be compared among treatments. The authors have added zero-time data indicating an initial VFA concentration of 20.3 mM. This is not “VFA produced” (as stated in the title), and presenting total (including carryover) rather than net VFA may obscure important differences. For example, the authors could not detect treatment differences at 6 h (29.0, 28.1, 26.2 mM) but had they analyzed net VFA (8.7, 7.8, 4.7 mM), they might have detected significant differences. By the same token, proper analysis of VFA molar proportions need to use net VFA production, not total VFA in the vial.

            In a few places (e.g., L347. L354), the authors have continued to use ‘increase” or “decrease” when comparing across treatments, instead of the more proper “were higher than” or “were lower than”.

            Throughout the manuscript, genus and species-level Latin names should be italicized (L266 et seq.), but family-level names (Table 4) should not be italicized.

Specific comments:

L280-298: The authors have added Pearson correlation data between VFA and bacterial taxa, and have provided correlation coefficients, but have not indicated which of these correlations are statistically significant.

L355-357: The reviewer does not follow the authors’ argument here. The differences in “organic acids” between cane and beet (Table 1) include aconitic and glycolic acid. What does this have to do with the activity or abundance of family Succinivibrionaceae?

L364-366: What it suggests to the reviewer is that the Ruminococcaceae do not use or compete for the sugars in molasses, and thus their overall relative abundance is lowered at the expense of those sugar fermenters whose relative abundance is higher.

L379-381: It is not clear how a positive correlation to butyrate and a negative correlation to acetate “reflects a preference …..for non-fibrous material”.

L381-389: The positive correlation with acetate and propionate reflect the fact that members of this genus are known to produce these acids, and the negative correlation with butyrate reflect the fact that Prevotella is not known to produce butyrate. This is independent of whether or not they degrade lignocellulose. And while some Prevotella can ferment xylan or other hemicelluloses, none are known to be truly cellulolytic.

Minor edits:

L54, L69: Change “by” to “of”

L67: Change “that” to “which”.

Table 1: Capitalize dextrans, “levans”; change “destrans” to “Dextrans”, and change “arabans” to “Arabinans”.

L181: Change “om” to “in”.

L266, L275: Change “genus” to “genera”.

L278: Change “as well as” to “as did”.

L282: Change “Fibrobact3riaceae” to “Fibrobacter” (the proper genus name).

L301: Change “already” to “previously”.

L313: Insert “numerically” ahead of “higher”. The P value of 0.11, though above the conventional P=0.05, still suggests the effect approaches statistical significance, and is worth noting.

L316: Change “an important substrate” to “important substrates”.

L317-318: Incomplete second sentence should be combined with the first sentence.

L345: Change “abundancy” to “abundance”.

L351: Change “taxa” to “taxon” (singular).

L377-378: Do not capitalize species names.

L398: Insert “relative” ahead of “abundance”.

Reviewer 2 Report

- Authors must include the results of the genus-level and correlation analyses in tables in the manuscript.

- The fact that this experiment was conducted in only 2 runs should be highlighted in the discussion section as a shortcoming of this study.

Round 3

Reviewer 1 Report

The reviewer is satisfied with the authors’ revisions. The reviewer wishes to point out a few additional editorials suggestions:

L212: Insert “total” ahead of “VFAs”.

L328: Change “concentrated” to “abundant”.

L329: Insert paragraph break here.

L334: Correct “bot” to “both”.

L356: Correct “eproduction” to “production”.

L366: Insert paragraph break here.

L405: Insert “the” ahead of “literature”.

Author Response

he reviewer is satisfied with the authors’ revisions. The reviewer wishes to point out a few additional editorials suggestions:

AU: authors thank the reviewer for his precious suggestions for the improvement of the whole manuscript

L212: Insert “total” ahead of “VFAs”.

AU: inserted

L328: Change “concentrated” to “abundant”.

AU: changed

L329: Insert paragraph break here.

AU: inserted

L334: Correct “bot” to “both”.

AU: changed

L356: Correct “eproduction” to “production”.

AU: changed

L366: Insert paragraph break here.

AU: inserted

L405: Insert “the” ahead of “literature”.

AU: inserted

Reviewer 2 Report

Authors have to pay more attention to the comments from reviewers to improve their own paper's quality, and you have to report the exact place where you made a correction; otherwise, this is time-wasting for both sides. The correlation analysis table is not yet included in the MS.

Author Response

Authors have to pay more attention to the comments from reviewers to improve their own paper's quality, and you have to report the exact place where you made a correction; otherwise, this is time-wasting for both sides. The correlation analysis table is not yet included in the MS. 

AU: authors are really sorry for the inconvenience, due to the automaic in-program revision system. The table is now included in the manuscript. Authors would also thank the reviewer for the corrections / suggestions made, which improved the quality of the paper.